# The Transient Nature of Emergent In-Context Learning in Transformers

**Aaditya K. Singh**[*]
Gatsby Unit, UCL

**Stephanie C.Y. Chan**[*]
Google DeepMind

**Ted Moskovitz**
Gatsby Unit, UCL

**Erin Grant**
Gatsby Unit & SWC, UCL

**Andrew M. Saxe**[†]
Gatsby Unit & SWC, UCL

**Felix Hill**[†]
Google DeepMind

## Abstract

Transformer neural networks can exhibit a surprising capacity for in-context learning (ICL) despite not being explicitly trained for it. Prior work has provided a deeper understanding of how ICL emerges in transformers, e.g., through the lens of mechanistic interpretability, Bayesian inference, or by examining the distributional properties of training data. However, in each of these cases, ICL is treated largely as a *persistent* phenomenon; namely, once ICL emerges, it is assumed to persist asymptotically. Here, we show that the emergence of ICL during transformer training is, in fact, often *transient*. We train transformers on synthetic data designed so that both ICL and in-weights learning (IWL) strategies can lead to correct predictions. We find that ICL first emerges, then disappears and gives way to IWL, all while the training loss decreases, indicating an asymptotic preference for IWL. The transient nature of ICL is observed in transformers across a range of model sizes and datasets, raising the question of how much to "overtrain" transformers when seeking compact, cheaper-to-run models. We find that L2 regularization may offer a path to more persistent ICL that removes the need for early stopping based on ICL-style validation tasks. Finally, we present initial evidence that ICL transience may be caused by competition between ICL and IWL circuits.

## 1 Introduction

In-context learning (ICL) is the ability of a model to use inputs at inference time to adapt its behavior, without weight updates, in order to solve tasks not present during training. The first examples of this behavior in neural networks were observed in architectures specifically designed and trained for *few-shot learning*, the capacity to learn a desired behavior from only a few examples [1–3].[2] Training in these cases involved shuffling the labels corresponding to input exemplars on each "episode" so that, to perform well on the training set, the model had to recall exemplar-label mappings from context to make future predictions. At test time, novel exemplar-label mappings were provided, and the network had to use these to classify query exemplars.

Research on ICL changed with the development of the transformer [5]. Brown et al. [6] first observed that a transformer-based language model, GPT-3, trained auto-regressively at sufficient scale, exhibited ICL without any specific effort of the authors to promote it via the training objective or data. Since then, a large body of literature has studied [7–12] or reported [13–15] examples of ICL in transformers trained at scale.

---

[*]Co-first authors; direct correspondence to `aaditya.singh.21@ucl.ac.uk` and `scychan@google.com`
[†]Co-senior authors
[2]Few-shot learning can also be implemented via methods that apply gradient updates [e.g., 4], but we focus on implementations of ICL that do not require any weight updates.

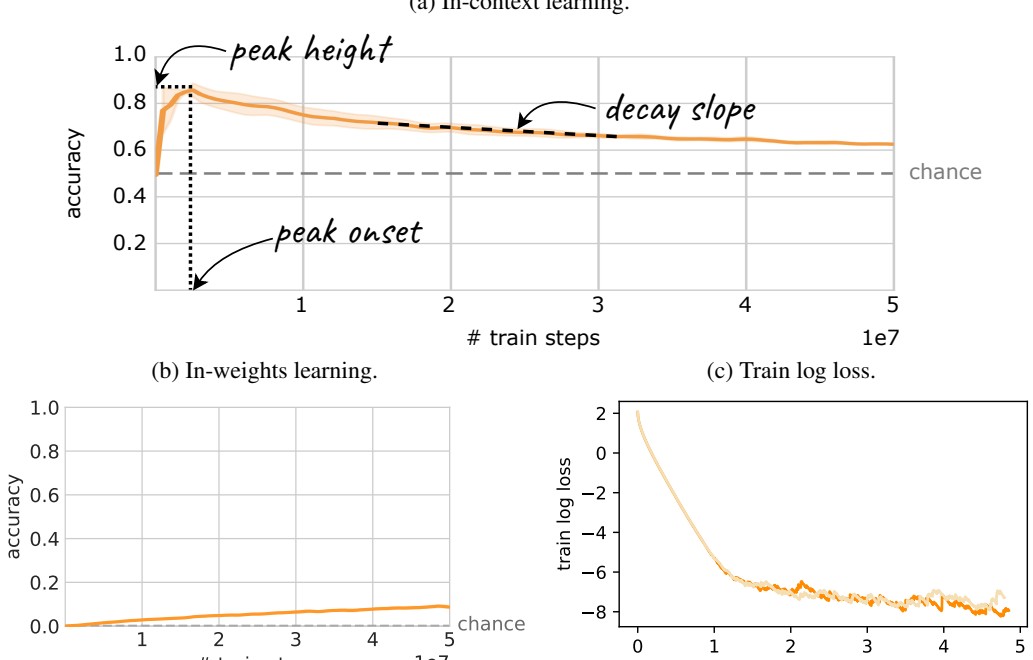

Figure 1: In-context learning is transient, shown for our "default" settings: 12 layers, embedding dimension of 64, trained on 1,600 classes, with 20 exemplars per class. All training sequences are bursty (see Figure 2a for details). Chan et al. [10] found these settings to strongly incentivize ICL, but did not observe ICL transience (see Figure 3), as they did not train long enough. (a) ICL evaluator accuracy. (b) IWL evaluator accuracy. We note that, while accuracy on train sequences is 100%, accuracy on the IWL evaluator is very slowly increasing, as the test sequences are out-of-distribution. See Appendix B for further investigation. (c) Training log loss. Two colors indicate two seeds used for experiments.

These compelling findings have sparked a surge of research into the phenomenon of emergent capabilities in large neural networks. Nonetheless, recent work has shown that ICL is not a *guaranteed* outcome of training transformers. Chan et al. [10] found that specific properties that are typical of language data, such as burstiness and its highly skewed distribution, play an important role in emergent ICL in transformers. When trained on data without these properties, Chan et al. [10] found that transformers tend to resort to *in-weights learning* (IWL). In the IWL regime, the transformer relies on information stored in the weights of the model, rather than new information provided in-context.[3] Importantly, ICL and IWL often appear to be in opposition, with ICL emerging most readily when training data has a large number of tokens or classes and when that data is bursty, with items appearing in clusters rather than being uniformly distributed.

Such controlled studies based on known data-generating distributions are critical for better understanding the ICL phenomenon in transformers. At the same time, a complementary body of work studies emergence in massive models trained directly on organic web-scale data, with the takeaway that impressive capabilities such as ICL are more likely to emerge in large models trained on more data [14, 17–19]. However, this reliance on massive models poses serious practical challenges, such as how to innovate rapidly, how to train energy-efficiently and in low-resource settings, and how to run efficiently at deployment. Thus, a growing line of research [20–22] has focused on achieving comparable performance (including emergent ICL) in smaller transformer models.

*Overtraining* is currently the favored approach for training small yet performant transformers. These small models are trained on more data, possibly with repetition, beyond what is prescribed by scaling laws and compute budget [13, 15]. At its core, overtraining relies on an assumption that is implicit in

---

[3]We also note that ICL and IWL are not so cleanly distinguishable in many cases: ICL can depend at least partially on information stored in-weights, and behavior that looks like IWL may still fact using context naively, e.g., via simple copying biases [16].

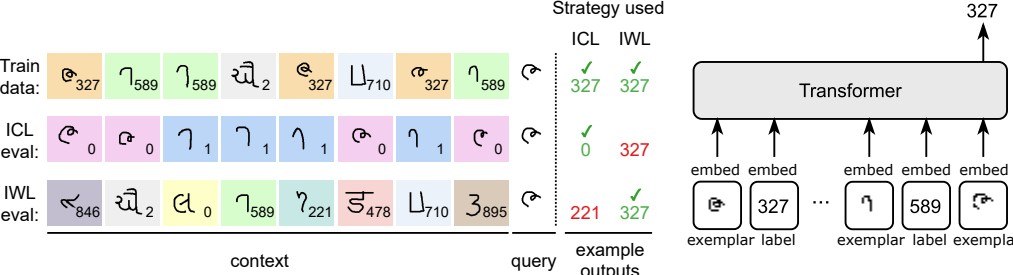

(a) Example sequences and outputs.     (b) Model schematic.

Figure 2: An overview of our setup. (a) Example sequences during training, ICL evaluation, and IWL evaluation. Example outputs are colored green when correct and red when incorrect. Note that for train sequences, ICL and IWL strategies *both* result in the correct answer. On ICL eval sequences, the IWL prediction is incorrect. ICL is required to remap exemplars to the randomized 0 or 1 labels. On IWL eval sequences, there are no matching exemplar-label pairs in context, so IWL is necessary. (b) Model schematic. Training and evaluation focuses on the predicted label for the final exemplar.

most (if not all) recent explorations of ICL in LLMs: *persistence*. That is, it is assumed that once a given model has been trained sufficiently for an ICL-dependent capacity to emerge, that capacity will be retained as training progresses, provided the training loss continues to decrease.

Here, we show that the general assumption of persistence is false. We do so by adapting a standard image-based few-shot dataset [23], which allows us to evaluate ICL carefully under controlled conditions. We characterize simple cases where ICL emerges, only to be subsequently lost while the model's loss continues to decrease. In other words, while ICL is well-known to be an emergent phenomenon, we should also think of it as a potentially *transient* effect (Section 3, Figure 1). We find this transience occurs across a range of model sizes (Section 4.1), dataset sizes (Section 4.2), and dataset types (Section 4.3), though we do identify some properties (Section 4.4) that can postpone transience. In Section 5, we study ways to alleviate this risk, and show that regularization may offer a path to persistent ICL. In Section 6, we present preliminary evidence that ICL transience is caused by competition with IWL circuits.

In general, if networks are trained carelessly for longer and longer, we find that ICL can disappear as readily as it emerges, leaving models without capabilities that we are increasingly expecting to find in modern AI systems.

## 2 Experimental Setup

### 2.1 Dataset construction

In order to study the transience of ICL across a range of model and dataset sizes in a controlled fashion, we use a synthetic data-generation process inspired by Chan et al. [10], which allows us to properly assess in-context *versus* in-weights learning. Our data generators are primarily built from the Omniglot dataset, a standard benchmark for few-shot learning, which consists of images of handwritten characters from different alphabets [23, MIT License]. The Omniglot dataset consists of 1623 character classes, with 20 exemplars each. In experiments with more than 1,600 classes, we construct additional image exemplars by applying rotations and flips of the original Omniglot images.

Inputs to the transformer model are sequences of exemplars and labels. We use 8 exemplar-label pairs (the "context") followed by one "query" exemplar, for an input sequence length of 17 (Figure 2a). The model is trained to minimize the loss in predicting the label for the query exemplar (Figure 2b).

To explore the generality of our results, we also extend to exemplars derived from language model token embeddings. We use pre-trained token embeddings from the LLaMa family [15] of large language models, which use a vocabulary size of 32,000 tokens. We extract the input embedding matrices from all four open-source LLaMa 1 models. We then subset and cluster [24] these token embeddings to form 3,200 classes of 5 tokens each. For each LLaMa model size, we construct two such datasets, with different clustering seeds. The full procedure is described in Appendix C.

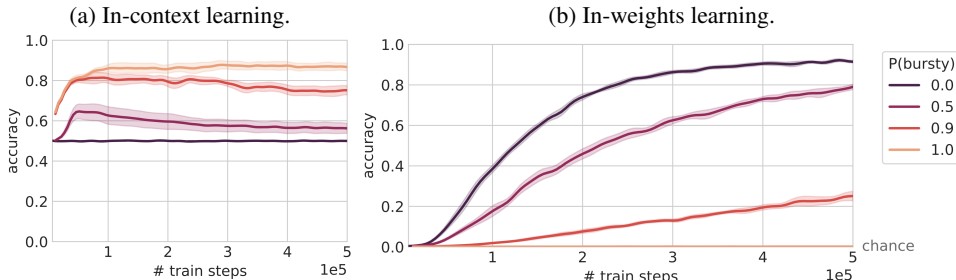

Figure 3: (Reproduced with permission from Chan et al. [10].) It was previously shown that ICL can be transient (purple and red curves) when transformers are trained on data that is only weakly conducive to ICL (e.g., with low levels of burstiness). P(bursty) indicates the fraction of training sequences that are bursty. All curves in this figure train on a dataset with 1,600 classes and 20 exemplars. Note the $x$-axis scale: At this number of iterations, there is no sign of ICL transience for the highest level of burstiness, but our experiments are run for a much larger number of steps. We also note that, when P(bursty) < 1, some of the sequences during training are of the same form as the IWL evaluation sequences. As a result, the IWL evaluator is less out-of-distribution, which is why the accuracies in (b), when P(bursty) < 1, are so high.

**Training sequences.** The training task (see top row in Figure 2a) can be solved via ICL *or* IWL. The context for each training sequence contains 3 exemplar-label pairs from the query class ("bursty" sequences), which prior work has found to incentivize ICL [10]). Bursty sequences allow the network to look for matching exemplars in-context to output a label for the query example, an example of ICL. To avoid repetition biases, the training context also includes 3 exemplar-label pairs from a distinct, distractor class. We keep exemplar-label mappings fixed across sequences, so the network could learn a mapping from exemplars to labels in-weights (without needing to find matching exemplars in-context). This design allows us to observe whether the model prefers ICL or IWL at different points in time, as a function of model size or in response to the properties of the training data.

**Evaluation sequences.** We follow Chan et al. [10] and construct two kinds of evaluation sequences that measure the extent to which a trained network relies on ICL *versus* IWL. To evaluate ICL (see second row of Figure 2a), we use sequences with 4 exemplar images from each of two classes, but we set the class labels to either 0 or 1 at random for each sequence (unlike during training, where the exemplar-label mappings are fixed). Accuracy on this evaluator is measured across 0 and 1 as possible outputs (so chance-level accuracy is 50%). As these labels were not associated with these exemplars during training, the only way to achieve above-chance accuracy on this evaluator is for the network to refer back to the context and copy the label corresponding to matching exemplars. To evaluate IWL (see third row of Figure 2a), we use sequences where none of the context exemplars come from the same class as the query, but all of the exemplar-label mappings are the same as during training. In this case, ICL is not useful (as there are no matching exemplars in-context), so the network must rely on exemplar-label mappings stored in-weights (IWL) for above-chance accuracy. Other evaluation techniques for ICL and IWL are considered in Appendix B.

## 2.2 Model details

For our transformer model, the "default" settings consist of 12 layers, with an embedding dimension of 64 and an additive, sinusoidal positional encoding scheme [5]. Each element of the sequence is passed through an embedder, either a standard embedding layer for the label tokens, a ResNet encoder for the exemplar images, or a simple linear layer for exemplars created from LLaMa tokens. (Figure 2b). Both embedders are trained jointly with the network. In our experiments, we vary multiple aspects of this architecture and observe the effect on the transience of ICL. All experiments were run with 2 seeds and used the Adam optimizer [25]. We did not observe qualitative differences in behavior between seeds. We ran most experiments for 5e7 iterations, but truncated some early when signs of transience were clear. More details (e.g., specific hyperparameter choices, and links to code) can be found in Appendix A.

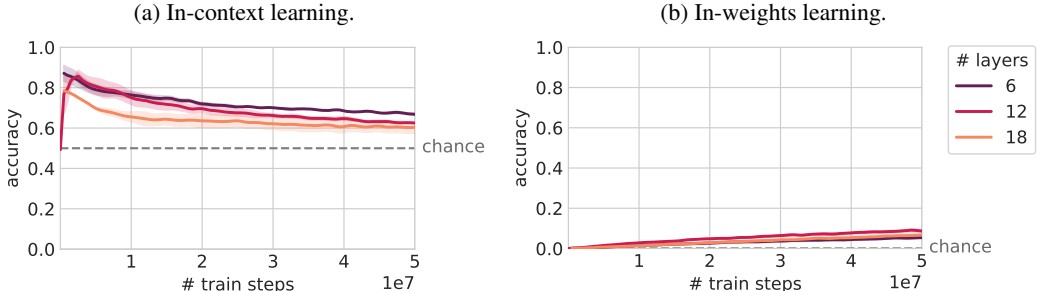

Figure 4: (a-b) ICL is transient regardless of model depth, with no clear trend of peak height or peak onset. Decay slopes are roughly similar across model sizes.

# 3 In-context learning is transient

Figure 1a shows that ICL is transient even when using data-distributional settings previously found by Chan et al. [10] to strongly incentivize ICL. We annotate various aspects of the ICL curve, such as the peak height, peak onset, and decay slope. Prior work [10] stopped training at 5e5 training steps, before peak onset. Training for much longer in an identical setting leads to the disappearance of ICL. Notably, ICL is slowly replaced by IWL (as seen by the steadily increasing accuracy on the IWL evaluator, Figure 1b), all while the training loss continues to decrease (Figure 1c).

Chan et al. [10] found some transient ICL in other data regimes, such as when some small percentage of sequences are non-bursty, where the context does not contain the query (Figure 3). In these settings, the eventual rise of IWL is to be expected, as ICL cannot minimize the loss on the non-bursty sequences. Notably, our results (Figure 1) show surprisingly that ICL transience extends even to cases where ICL is "sufficient" to fully solve the training task (such as when the context always contains a relevant exemplar-label mapping that can be used). When comparing to the results from Chan et al. [10] (Figure 3), we see that higher levels of burstiness lead to a higher peak height and gentler decay slope. When training for fewer iterations, it might appear that ICL is *persistent*, but our results from training longer in Figure 1 demonstrate that ICL is actually *transient*.

# 4 In-context learning is transient across settings

## 4.1 Effect of model size

We investigate whether the ICL transience effect is modulated by model size, given the current prevalence of extremely large transformer models. We find that there is no consistent effect of model depth on ICL transience (Figure 4a-b). In fact, deeper models can lead to lower peak heights (e.g., the 18-layer model has a lower peak height than the 12-layer model). Notably, the decay slope is similar across different model depths, indicating that the ICL is unlikely to persist simply by scaling model depth.

We also experiment with the width of the model in Section 6, which sheds some light on the causes of ICL transience.

## 4.2 Scaling dataset size

Next, we investigate the effects of dataset size on ICL transience. There are two primary ways of varying the dataset size: increasing the number of exemplars per class (a form of increasing in-class variation), or increasing the number of classes.

Chan et al. [10] found that in-class variation was crucial for ICL to emerge, and we find a similar trend. When the number of exemplars per class is increased (from 10 to 15 to 20) while keeping the number of classes fixed, we see an increase in ICL peak height (Figure 5a). However, despite higher peak heights, we also see steeper decay slopes, indicating that increasing in-class variation is unlikely to make ICL persistent.

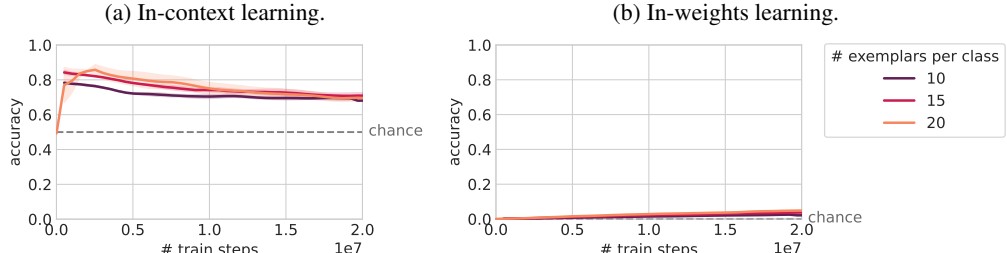

Figure 5: In-class variation improves ICL, as previously known and seen here by the higher peak heights, but ICL is nonetheless transient across settings.

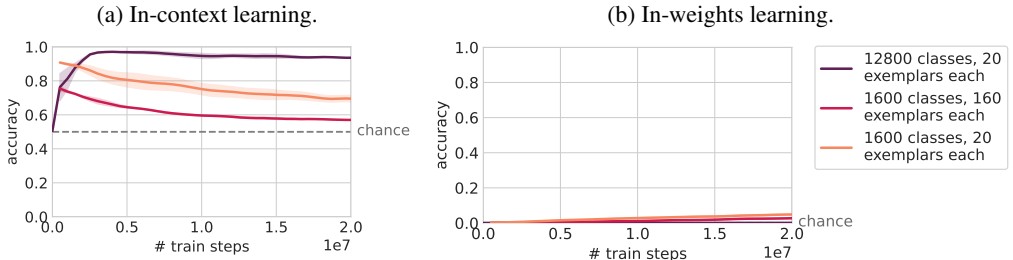

Figure 6: Effect of dataset size on ICL transience. Increasing the number of classes leads to higher ICL peak height and a more gradual decay slope. Despite the number of exemplar-label pairs being the same in the purple and red curves, we note that the ICL and IWL behavior is very different when we increase the number of classes (purple) *versus* the in-class variation (red). Also, note that chance level performance for IWL differs slightly: 1/12,800 for the other curve (purple), and 1/1,600 for the 1,600 classes lines (red and orange).

When increasing the number of classes, we again find a similar trend to Chan et al. [10] with peak heights increasing (Figure 6a). In this case, we also find that decay slopes become more gradual. To better illustrate the differences between increasing in-class variation and increasing the number of classes, we do another experiment where the total number of exemplar-label pairs is fixed to $12,800 \times 20 = 1,600 \times 160 = 256,000$, with one run having more classes and less in-class variation, and another run having fewer classes and more in-class variation. We find that increasing the number of classes is a more effective way to promote ICL (higher peak height, shallower decay slope) than increasing the in-class variation, perhaps due to each class appearing less often during training. This result also indicates that, beyond a certain point, in-class variation actually hurts ICL performance (lower peak height, steeper decay slope, as seen in Figure 6a). To conclude, ICL remains transient as we increase the number of classes, but training for longer and longer becomes necessary to make it go away, given the shallower decay slope.

## 4.3 Extending to language model token embeddings

To explore the generality of our results, we replaced the image exemplars with exemplars derived from language-model token embeddings (procedure described fully in Appendix C). On all of these

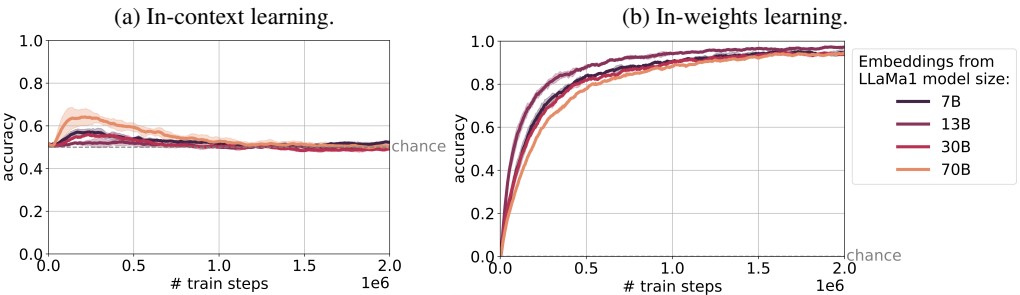

Figure 7: ICL is transient across a range of input datasets, derived from clusterings of LLaMa 1 token embeddings from the four different sizes. Error bars indicate standard error across two possible datasets derived from each model's embedding matrix. Further results can be found in Appendix C.

token-embedding derived datasets, we find that whenever ICL emerges, it is subsequently transient (Figure 7). Furthermore, the peak height is lower than in our other experiments. In terms of timing, the peak onset as well as subsequent transience occurs on a quicker timescale (note the shorter $x$-axis scale in Figure 7). These results also further corroborate our findings on dataset size (Section 4.2), as the constructed token embedding-label dataset is smaller ($3,200 \times 5 = 16,000$ exemplar-label pairs, as opposed to $1,600 \times 20 = 32,000$ pairs in our Omniglot experiments), so we would expect lower heights and sharper decay slopes. Overall, these experiments are a promising step towards connecting our findings to natural language, and demonstrate that the transience phenomenon is not limited to image-based exemplar-label datasets.

### 4.4 Changing the data distribution

Until now, we have trained on data for which the marginal distribution over classes was uniform. However, a common property of naturalistic data is a Zipfian (power-law) skew [26]. Zipfian data contains a few "common" classes that appear very often, and many "rare" classes that appear less often than they would if the class distribution were uniform. In Section 4.2, we found that more classes (corresponding to each class appearing less often) led to shallower decay slopes. We explored whether data with a Zipfian distribution could play a similar role in reducing the transience of ICL.

Indeed, in Figure 8, we find this to be the case. We implemented different levels of skew across data classes according to a Zipfian distribution $p(X = x) \propto x^{-\alpha}$ ($x$ corresponds to the rank of an input class—e.g., $\alpha = 0$ corresponds to a uniform distribution). At Zipf exponent $\alpha = 1$, we find a severely delayed ICL peak onset, thereby delaying the subsequent transience, and a shallow decay slope following this delayed onset (similar to that seen with a large number of classes in Figure 6a). Furthermore, corroborating the findings of Chan et al. [10], we find that stopping training early, when using skewed distributions, can lead to a combination of ICL and IWL in the same network (as evidenced by the increased IWL at Zipf exponent $\alpha = 1$ as compared to Zipf exponent $\alpha = 0$, Figure 8b). However, it is important to note that if the Zipfian exponent is too extreme (e.g., $\alpha = 2$), ICL does not emerge at all, and the network uses IWL to solve the task. We conclude that moderately skewed Zipfian data does not eliminate transience, but mitigates it by necessitating training for an even longer time to observe transience (similar to when there is a large number of classes).

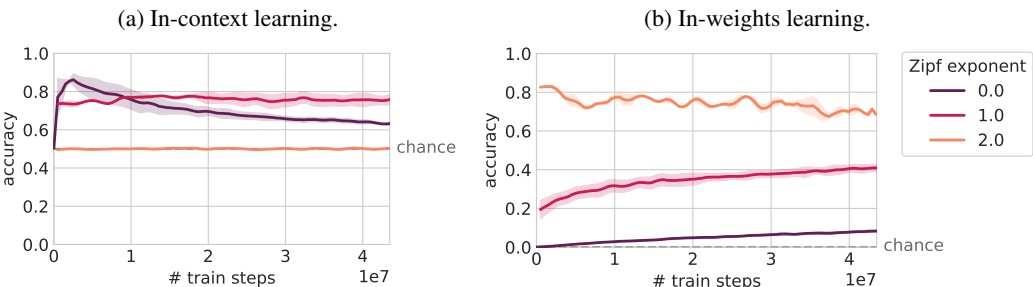

Figure 8: ICL transience as a function of dataset skew, implemented across classes in accordance with a Zipfian distribution. (a) A moderate Zipf exponent of 1 leads to a significant delay of ICL peak onset followed by a shallow decay slope, implying models would need to be trained for even longer to get full transience. However, an extreme level of skew (Zipf exponent 2) leads to the complete loss of ICL altogether. (b) IWL, evaluated on the same skewed distribution over query classes as in the training data (thus high accuracy on Zipf exponent 2 is not unexpected, as the IWL test set is dominated by a small number of classes).

## 5 Regularization may eliminate ICL transience

If we consider that ICL is a more generally useful solution than IWL, then ICL transience is, in a sense, the *opposite* of the grokking phenomenon, where a model changes from a less to a more general solution after long periods of training [27]. Regularization (or implicit regularization) may help to drive grokking [28, 29] (and some work even shows an impact of dataset size as well [30], analogous to what we find above). Thus, would we see regularization mitigating the ICL transience phenomenon?

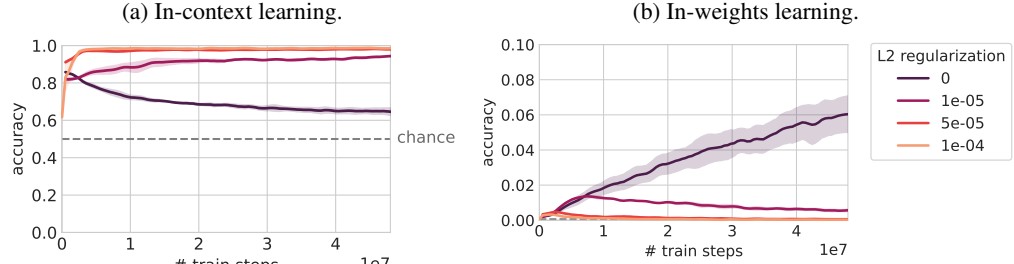

Figure 9: L2 regularization potentially eliminates ICL transience. Increasing regularization decreases ICL transience, at least up to a point. Regularization also seems to lead instead to IWL transience. (Note that the $y$-axis in Figure 9b is zoomed in so that IWL transience is visible.)

We see in Figure 9 that this is indeed the case, with L2 regularization seeming to even eliminate ICL transience entirely, at least on the timescales that we tested. Without any regularization, we see transient ICL as before. However, as we add increasing levels of L2 regularization, we see that the decay slope of ICL performance goes to 0. On the other hand, we actually observe some transience of in-weights learning (purple and maroon curves, Figure 9b). These results indicate that the ICL circuit is a lower norm solution than the IWL circuit. Thus, regularization may offer a path to persistent ICL. However, with values of regularization that are too high ($> 0.0001$), the network is over-regularized and ICL transience returns, but, notably, IWL accuracy never rises above zero either (see Appendix D for more train loss curves and more extreme values of regularization).

## 6 ICL transience may be caused by competition with IWL circuits

We performed initial investigations to understand the origins of ICL transience: why does ICL fade when the IWL solution emerges? Do the ICL and IWL circuits compete with each other for resources, for example, in the transformer's residual stream [16]? If so, we would expect wider transformers with larger embedding size (and therefore greater residual stream capacity) to be less afflicted by ICL transience. In Figure 10a, we see that this is the case: Greater model embedding sizes lead to higher peak heights and gentler decay slopes. However, one confound to consider is that larger embedding may just independently enhance ICL abilities (as they enhance IWL abilities, Figure 10b), rather than reduce competition between circuits.

To better understand how scaling the embedding dimension enhances ICL and IWL, we trained models on data that could be solved via ICL only or IWL only, rather than being solvable by both strategies. For ICL-only training, we used the same type of sequences used in our ICL evaluator, where the network *must use ICL* to solve the query-prediction task. For IWL-only training, we used the same type of sequences used in our IWL evaluator, where the network *must use IWL* to solve the query-prediction task. Figure 10d shows that larger embedding dimensions enhance the model's ability to perform IWL, but do not consistently help ICL (Figure 10c). Thus, these results indicate that the increased peak height and shallower decay slope in Figure 10a are likely due to decreased competitive interactions with IWL circuits, rather than an enhancement of ICL capabilities.

We arrived at convergent evidence by building on the regularization experiments from Section 5. We applied weight decay selectively to either the ResNet embedding layers, the MLP layers, or the self-attention layers. When we applied weight decay to only the self-attention layers, ICL was still transient. When we applied weight decay to the MLP or ResNet layers, however, ICL transience was mitigated (Appendix Figure 16). These results can be interpreted in light of prior work indicating that in-weights information is stored in MLP layers [31, 32]. By selectively penalizing this behavior, we enable ICL to persist, thus providing convergent evidence that – when no weight decay is applied – ICL fades due to competition with IWL circuits.

## 7 Related Work

This work builds on several papers that seek to understand in-context learning by analyzing small networks and/or controlled data sources. Xie et al. [7] consider an implicit definition of ICL (defined in terms of the model's loss) and show that the effect can be framed as a form of Bayesian inference. Chan et al. [10] show via experimentation with controlled synthetic datasets that, e.g., the bursty and

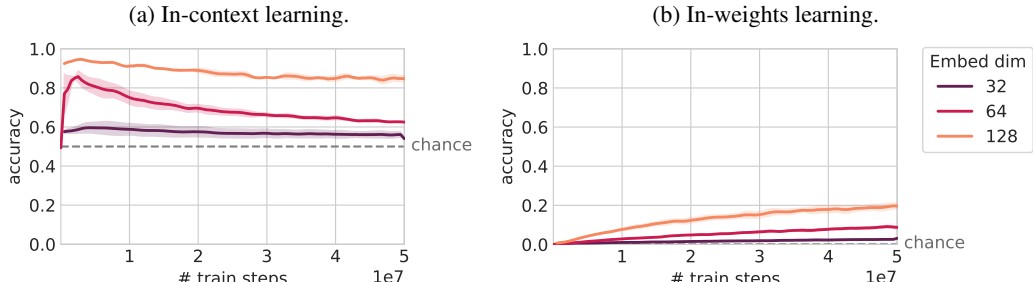

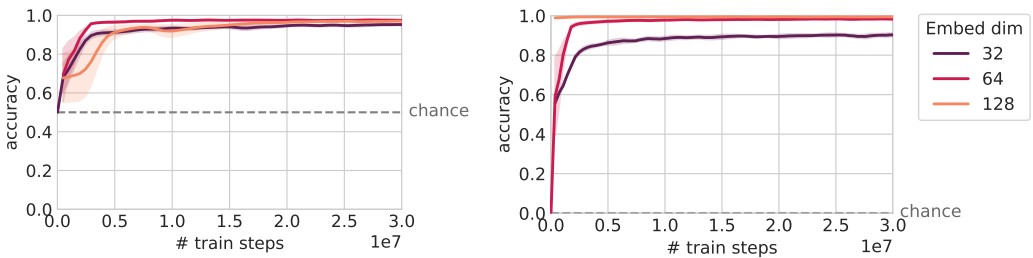

Figure 10: (a-b) We find a trend that wider models (larger embedding sizes) have higher ICL peak heights and more gradual decay slopes. (c-d) However, when we trained on data that allowed us to specifically elicit each type of learning, we found that embedding size was only consistently helpful for enhancing IWL, indicating that mitigation of ICL transience is likely due to decreased interaction between the two circuits, as opposed to an enhancement of ICL capabilities.

Zipfian nature of text data could play an important role in the emergence of ICL. Many other works consider how ICL can be used to learn certain function classes using synthetic data, and how ICL may be implementing a form of gradient descent to accomplish this [33–35].

As with the aforementioned work, here we do not experiment directly with billion-parameter transformers trained on web-scale corpora. Nonetheless, our work should be of relevance to studies that aim to derive ICL in increasingly small and compact text-based language models. By training language models of many different sizes on different amounts of data, Hoffmann et al. [13] identified a relationship between the amount of training and model size that appeared compute-optimal with respect to the training loss. This correspondence motivated Chinchilla, a 70B-parameter language model whose downstream performance (on tasks that require ICL) was comparable to far larger models. More recently, Touvron et al. [15] took this further, "overtraining" smaller models to achieve lower training loss according to a given, inference-time, compute budget. Perez et al. [36] point out that in such work, tasks that require ICL are typically included in the models' validation set in order to optimise hyperparameters directly for ICL. Our results indicate that, without appropriate validation and early stopping, training for a long time may eventually lead to ICL gradually disappearing, perhaps even without recognition from model developers. A large number of "classes" or Zipfian data distributions (both properties of natural language [37]) can mitigate this by delaying the disappearance. Moreover, regularization may completely eliminate this risk. Indeed, Hoffmann et al. [13] find that the use of optimizers with weight decay [38] leads to models that perform better, but this benefit only manifests at later stages of training (perhaps when ICL would otherwise decay).

## 8   Discussion

**In-context learning is often transient.** The key insight of this work is the empirical finding that, if in-context learning (ICL) emerges in a transformer network, it may not necessarily persist as the model continues to be trained. That is, while ICL is often emergent, it may be in many cases also be transient (Section 3). In fact, since the loss of ICL is gradual, its creeping disappearance may not even be known to model developers. This emphasizes the importance of evaluating appropriate validation metrics over the course of training [e.g., 39]. Furthermore, these results suggest that it may be desirable to stop training some transformer models before a compute budget is reached—to

preserve ICL—even if training losses are still decreasing. This would need to be ascertained on a case-by-case basis. Other methods for preventing the disappearance of ICL, like regularization, may also be critical for delivering flexible, in-context learners.

**What are strategies for mitigating ICL transience?** In Section 4, we found that data distributional properties (increasing the number of classes, adding a Zipfian skew) can strengthen ICL and *postpone* the transience of ICL, and so can increasing embedding size (and least in the ranges we tested). The fact that most transformers are trained on language, which naturally has both of these data-distributional properties [37], may explain why ICL transience hasn't been documented in LLMs yet. However, as the trend of overtraining continues, there's a risk of losing these impressive ICL capabilities. We found that the most robust method to make ICL persistent was L2 regularization (Section 5).

**Why does ICL emerge at all, if it is not asymptotically preferred?** Since GPT-3 [6], the emergence of ICL has become an expected phenomenon in large-scale transformer models. Our experiments underline how curious it is that ICL emerges in such models given that it does not seem to be asymptotically preferred. Possible explanations may concern optimization quirks (similar to the slingshot mechanism proposed as an explanation for grokking [29]) or perhaps lottery tickets contained in common initialization methods [28, 40] that incentivize ICL (when trained with relevant data-distributional properties). Future work could explore these possible explanations by considering the effect of alternate optimizers or initialization schemes on the transience of ICL.

**Given that ICL does emerge, why does it fade in favor of IWL?** On our training data, either ICL or IWL can reach good performance. While we do not know for certain why ICL emerges in a network, the question remains of why it fades if it is a viable strategy to solve the training task. Our results in Section 6 suggest that part of the explanation may lie in a competition between IWL and ICL circuits for resources in the transformer residual stream [16]. Under the assumption that IWL is asymptotically preferred, this competition would explain why ICL fades after emerging.

**Why is IWL asymptotically preferred over ICL, when both solve the task?** Induction heads [9] are a possible circuit that may be responsible for ICL, and comprise of two interacting components: finding a match in-context, then copying some token forward. The match operation is fundamentally limited by the softness of the attention operation [41–43]. On the other hand, IWL relies on learning the exemplar-label mapping in-weights, which we show is feasible in Figure 10d. We postulate—and hope to investigate in future work—the possibility that although ICL and IWL can achieve perfect accuracy on the task, this "imperfect match" to prior context asymptotically incentivizes solutions (when training with cross-entropy loss, as is standard) that do not rely as much on context and instead learn in-weights.

**How do we get ICL and IWL to co-exist asymptotically in the same model?** Large language models clearly show a co-existence of ICL and IWL. Chan et al. [10] found that training on Zipfian data allows the network to perform ICL on rare classes, but learn common classes in-weights. Regularization was the best strategy we found to make ICL persistent, but it leads to little-or-no IWL. Given our results, the only effective way currently to get co-existence of ICL and IWL appears to be stopping training at the right time. Future work could investigate other factors that may restore co-existent ICL and IWL, even asymptotically.

**Limitations and future directions.** Perhaps the most significant limitation is that our experiments were not performed on large language models, one of the most important applications of transformers. Our training resources limited our ability to perform such experiments in the current work, but we believe that this is a necessary future step, to understand the full import of the current results. As a first step toward this, we demonstrated ICL transience on datasets with language model token embeddings (Section 4.3), but our datasets are still restricted to the exemplar-label setup. It would also be interesting to explore the effect of context length: For a given train or inference budget, a system designer could trade off between longer context (possibly enabling better ICL) and a larger model (possibly enabling better IWL). We also note that our definitions of ICL and IWL are relatively specific, and many more important mechanisms are likely at play in transformers—for example, contextual information could be used in conjunction with IWL to narrow down possible outputs via copying [16], which we find some preliminary evidence of in Appendix B. Future work could investigate more mechanistic explanations of ICL transience, the tradeoff between ICL and IWL, and additional recipes for reducing ICL transience.

## Acknowledgments and Disclosure of Funding

A.K.S. and T.M. are supported by the Gatsby Charitable Foundation. This work was supported by a Schmidt Science Polymath Award to A.S., and the Sainsbury Wellcome Centre Core Grant from Wellcome (219627/Z/19/Z) and the Gatsby Charitable Foundation (GAT3755). A.S. is a CIFAR Azrieli Global Scholar in the Learning in Machines & Brains program.

The authors thank Kira Düsterwald and DJ Strouse for insightful discussions throughout the course of the project. In addition, the authors thank Andrew Lampinen, Pierre Richemond, Roma Patel, and Murray Shanahan for feedback on the draft.

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

# A  Model and training procedure details

All experiments used the same model and training procedure unless stated otherwise. The transformer [5] consisted of 12 layers with am embedding dimension of 64 and 8 heads. The images were embedded by a ResNet [44] with two blocks per group and channels per group (16, 32, 32, 64). The ResNet embedder was learnt jointly and not pre-trained. The integer labels were embedded using a standard embedding layer. The input embeddings were augmented with a standard sinusoidal positional encoding [5] with timescale 30 (as our sequences are all length 17). Experiments were run for up to 5e7 training steps with batch size 32, on 16 TPU v2 or v3 cores. They were trained using Adam [25] (with default parameters of $\beta_1 = 0.9, \beta_2 = 0.999$) and a learning rate schedule with a linear warmup up to a maximum learning rate of 3e-4 at 4,000 steps, followed by an inverse square root decay. L2 regularization was implemented by adding the squared weights of the model (excluding batch norm parameters) to the loss term. All experiments were run with 2 seeds each. In all figures, (shaded) error bars indicate standard deviation around the mean.

Also, we note that, despite the long training times, all our experiments are still in the <1 epoch regime, due to the combinatorial nature of our dataset. We estimate there are over $1,600 \times 1,599 \times (1,598 \times 1,597/2) \times (8!/(3! \times 3!)) \times (20^9) = 1.87 \times 10^{27}$ possible training sequences, of which our model sees a maximum of $32 \times 5 \times 10^7 = 6.4 \times 10^8$.

Source code can be found at `github.com/google-deepmind/emergent_in_context_learning` and `github.com/aadityasingh/icl-transience`.

# B  Alternate evaluation metrics

While the primary evaluations used throughout the paper are inspired by the ICL and IWL evaluators of Chan et al. [10], we did consider a few alternate evaluators. Prior work has shown that transformers are prone to simple copying biases [16] and that the importance of few-shot exemplars might be more in their format rather than correctness [8, 11]. To account for copying biases, we introduce the "IWL-copy-available" evaluator, wherein we randomly replace one of the labels with the true label. Note that, during training, the correct label always appears in context, which may incentivize the formation of such naive copying circuits. To perform a stricter test of ICL performance, we thus take inspiration from the "flipped-ICL" evaluation in Wei et al. [11].

For clarity, we illustrate some abbreviated example sequences below:

- Train: X 23 Y 24 X 23 X 23 A 0 Y 24 B 1 Y 24 X ?. Note there are 3 instances of X 23, 3 instances of Y 24, and one instance each of A 0, B 1. Exemplar X is always associated to label 23 through training, Y to 24, A to 0, and B to 1.
- IWL eval (in main text): A 0 C 2 F 5 B 1 Y 24 D 3 H 7 X ?. Niether the exemplar nor label appears in context.
- IWL-copy-available eval: A 0 C 2 F 5 B **23** Y 24 D 3 H 7 X ?. We randomly replace one of the labels with the desired answer label. Note, this task allows the network to benefit from copying, but still provides no information *in-context* to learn that X should map to 23.
- ICL eval (in main text): X 0 X 0 Y 1 X 0 Y 1 Y 1 X 0 Y 1 X ?. Both X and Y's labels are replaces with 0/1. Note, we also tried using other pairs of labels for the ICL evaluator (e.g., 15/20), but found little variance across this choice.
- flipped-ICL eval: X 24 X 24 Y 23 X 24 Y 23 Y 23 X 24 Y 23 X ?. Note, if the network outputs 24 here, it demonstrates in-context learning, as it must be using the exemplar-label from context. Alternatively, if the network outputs 23, it demonstrates in-weights learning, as its ignoring the exemplar-label mapping provided in context. Accuracy=1 on this evaluator means the network is doing ICL, Accuracy=0 means the network is doing IWL.

Following the results of Section 4.3, which showed that fixed embeddings exhibit transience on shorter timescales, we conducted some experiments using the above evaluators on a dataset of fixed omniglot embeddings by preprocessing all images through a Imagenet-pretrained-and-frozen Resnet18 encoder [44, 45]. Like the main paper, we considered datasets with either 1,600 or 12,800 classes, with 20 exemplars per class. We did not run these experiments as long as those in the main paper, as the purpose was to test the new evaluators.

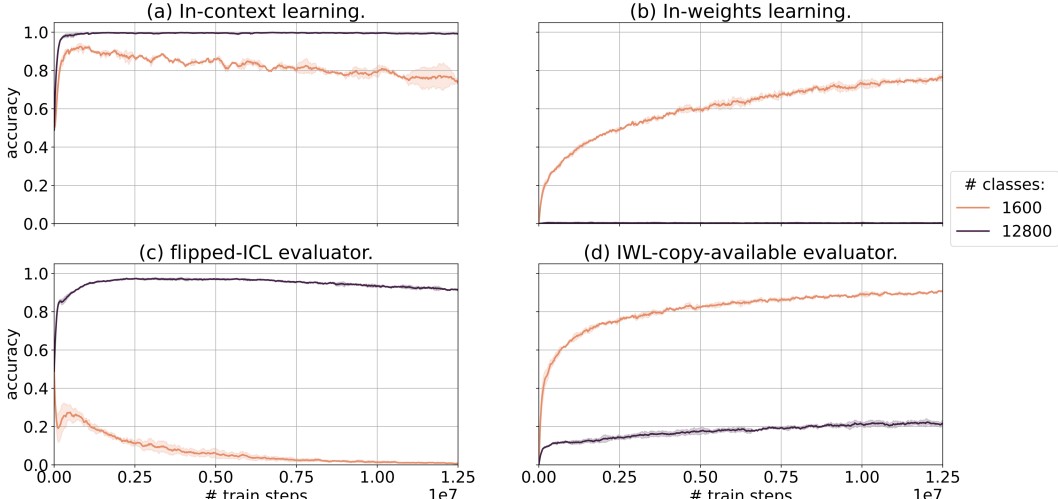

Figure 11: Results of alternate evaluators on fixed embedding Omniglot data. Note the shorter $x$-axis scale. (a) ICL evaluator, same as main text. (b) IWL evaluator, same as main text. (c) flipped-ICL evaluator. Note how signs of transience are visible sooner on this stricter measure. (d) IWL-copy-available evaluator. Similarly, IWL-copy-available starts growing sooner than the main IWL evaluator, offering earlier indications of transience

As seen in Figure 11, the alternate evaluators' asymptotic behaviors largely mirror the evaluators used by Chan et al. [10]. That said, there are some interesting differences. For example, under the flipped-ICL evaluator, models trained on 1,600 classes are never able to fully override the mapping of exemplars to labels seen in training (as evidenced by accuracy remaining below 0.5). However, at the scale of 12,800 classes, models are able to (accuracy goes above 0.5 before decaying). Intriguingly, these results may connect to Wei et al. [11], except in the sense of a more diverse dataset rather than a larger parameter count, leading a model to "do in-context learning differently".

With respect to transience, we do still observe it in both cases. For 1,600 classes, the increase in performance indicates the network may be overriding its exemplar-label mapping on some sequences, but this fledgling ability is short-lived. For 12,800 classes, while the shallow decay slope makes transience on the standard ICL evaluator not visible due to the short timescale (note, ICL is still transient at this # of classes, as shown in Figure 5a), transience is clearly visible on the flipped-ICL eval.

Furthermore, we see that there is indeed an effect of copying in our networks, as seen by the improved performance of models on the IWL-copy-available evaluator as compared to the IWL evaluator.

For a clearer comparison to prior work on synthetic datasets, we opted to use the same evaluators as Chan et al. [10] in this work, but hope the above analysis shows the importance of good evaluators in diagnosing ICL transience.

## C  Construction of LLaMa-based exemplar-label datasets

In this section, we provide more detail as to the procedure used to construct the fixed embedding datasets used in Section 4.3.

### C.1  Constructing datasets

We save the token embedding matrix for each LLaMa v1 model. These have dimensions 32,000 $\times d_{model}$, where $d_{model} = 4096, 5120, 6656, 8192$ for the 7B, 13B, 30B, and 70B, LLaMa models, respectively. To go from these raw embeddings to "classes" that we can use in our setup, we perform the following steps:

1. Subselect: Many tokens correspond to individual bytes or characters from non-English languages. We observed that the statistics of these token embeddings are often different than

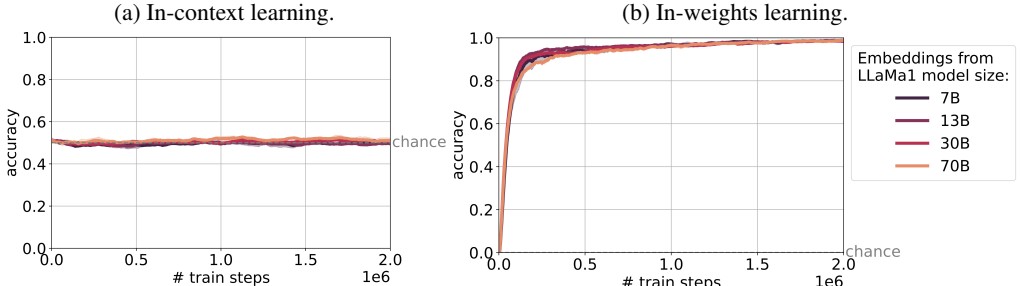
Figure 12: A model initialization seed for which no ICL emerges on LLaMa-derived datasets

the others, so we remove these tokens. Specifically, we use tokens 259 to 29,870, inclusive, for a total of 29,612 tokens.

2. Cluster: We cluster the token embeddings using K-means clustering and cosine distance using the FAISS library [24]. As FAISS does not offer a valid minimum number of points per cluster, we first cluster into 2400 (or 4800) clusters, then pick all clusters that have more than 10 (or 5) points. This gives us at least 1,600 (or 3,200) clusters. We randomly pick the actual 1,600 (or 3,200) clusters.

3. Select from cluster: Many of these clusters contain more than 10 (or 5) points. To maximize in-class variation, we pick the furthest 10 (or 5) points from the cluster centroid. Thus, we have our 1,600 classes of 10 examples, or 3,200 classes of 5 examples.

Note that step 2 is highly dependent on the random seed used to initialize cluster centroids. We run the above procedure for all 4 LLaMa models on 2 random seeds.

## C.2   Example clusters

If we map the embeddings back to tokens, we can get a sense of what various clusters may represent. Many clusters cover similar tokens, but some clusters can be more semantic or even multilingual! We show some extracted clusters (for 3,200 classes, 5 embeddings per class, on the 70B LLaMa) below:

```
_erm | _room | rm | _Room | _rm
_utter | _extremely | _partially | _entirely | Comple
hover | _mysqli | gz | RGB | rgb
async | _predicate | Observer | _deleg | _delegate
ive | ous | _cable | rable | ible
FLAG | _flag | _Flag | flag | Flag
_specified | _provided | _supplies | _supply | _supplied
ring | _dent | dent | _rent | RENT
_Derby | _Dit | _pedig | _ort | _Ort
_Britain | _England | _Madrid | _Spanish | Espagne
```

## C.3   The difficulty of emergent ICL

We observed that ICL emergence was more sensitive to model initialization when using LLaMa token embeddings as exemplars. An example seed where ICL did not emerge at all (for 3,200 classes, 5 exemplars per class) is shown in Figure 12. However, in all cases where ICL did emerge (even to a small degree), we found it to be transient (Figures 7, 13). Furthermore, the results presented in Section 4.3 are averaged over two different clustering seeds (see step 2 above), which yield different output datasets. ICL is transient across all 8 datasets tested.

At 1,600 classes, with 10 exemplars per class, we found mild ICL emergence, as shown in Figure 13. We found that 3,200 classes were necessary to get any strong ICL, which is shown in Figure 7.

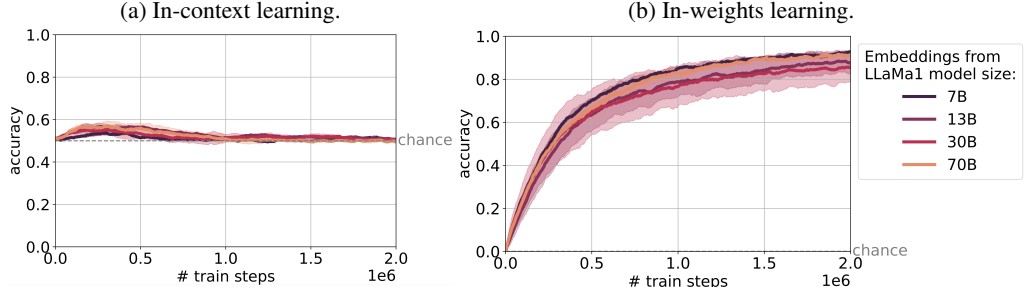

Figure 13: Mild emergence and subsequent transience of ICL on a model trained on the LLaMa derived datasets with 1,600 classes and 10 exemplars per class.

# D    Additional regularization results

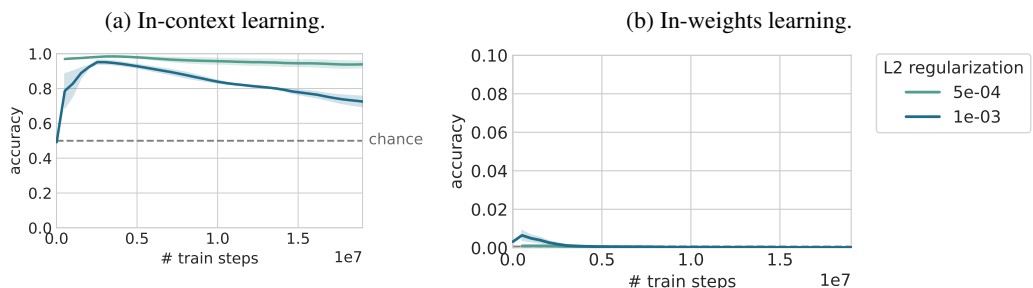

Figure 14: For extreme values of L2 regularization (>1e-04), the model seems to be over-regularized: We observe a return to ICL transience, as well as poor performance on IWL.

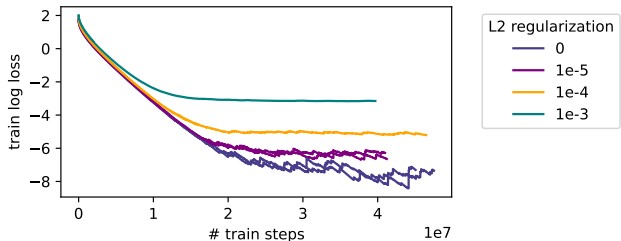

Figure 15: Total train loss (including L2 penalty) for a sampling of experiments. Regularization seems to make the training loss smoother, as well as causing train loss to plateau at a higher value (rather than continuing to drop).

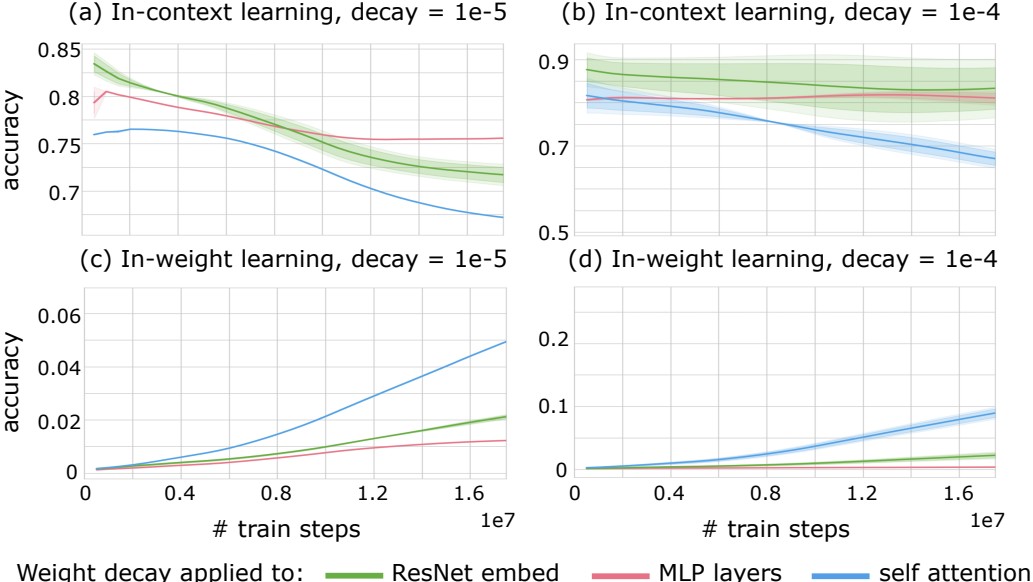

Figure 16: Selectively applying weight decay to different sets of network weights reveals that IWL relies on MLP layers. When we apply weight decay only to self-attention layers (blue), ICL is still transient. When we only apply weight decay to MLP layers, ICL transience is mitigated (red). These results can be interpreted in light of prior work indicating that in-weights information is stored in MLP layers [31, 32]. By selectively penalizing this behavior, we enable ICL to persist, thus providing convergent evidence that – when no weight decay is applied – ICL fades due to competition with IWL circuits.

