# OpenReview forum: "The Transient Nature of Emergent In-Context Learning in Transformers"
_NeurIPS.cc/2023/Conference — NeurIPS 2023 poster_

### Official Review · Reviewer_8T18 · 2023-07-03

**Soundness:** 3 good
**Presentation:** 3 good
**Contribution:** 3 good
**Rating:** 7
**Confidence:** 4

**Summary:**

The paper explores the in-context learning (ICL) phenomenon in transformer networks and investigates its transience during training. It presents empirical findings that ICL if emerged in transformers, may not persist as the model continues to be trained.
The paper highlights the importance of evaluating appropriate validation metrics and suggests stopping training before a computing budget is reached to preserve ICL.
Strategies to mitigate ICL transience, such as data distributional properties and regularization, are explored. The paper also discusses the competition between ICL and in-weight learning (IWL) circuits and raises questions about the emergence and fading of ICL.
The experiments demonstrate that ICL in transformers is often transient. Model size, dataset size, and data distribution can influence the transience of ICL, but persistent ICL is not guaranteed. Wider models with larger embedding sizes show improved ICL, possibly due to reduced competition with IWL circuits. Skewed data distributions mitigate ICL transience but necessitate longer training.
Future directions, limitations, and implications for large language models are discussed. Overall, the paper provides valuable insights into the nature of ICL and its implications for transformer networks, contributing to the understanding of in-context learning and its behavior during training.

**Strengths:**

Originality and Contribution:
- Empirical findings on the transience of in-context learning (ICL): The paper presents novel empirical evidence that ICL if emerged in transformer networks, may not persist as the model continues to be trained. This finding sheds light on an important characteristic of transformer models and highlights the need for appropriate evaluation and validation metrics.
- It provides insights into how ICL emerges, disappears, and gives way to in-weight learning (IWL) during training.
- Discussion of the coexistence of ICL and in-weights learning (IWL): The paper explores the competition between ICL and IWL circuits for resources in the transformer residual stream. It raises intriguing questions about the emergence and fading of ICL and offers potential explanations for the coexistence of ICL and IWL in large language models.
- Exploration of strategies to mitigate ICL transience: The paper investigates various strategies to mitigate the transience of ICL, including increasing the number of classes, introducing Zipfian data distribution, and using L2 regularization. These strategies provide valuable insights into potential methods for preserving and enhancing ICL capabilities.
- It is interesting that ICL is transient even under conditions that are deemed conducive to the emergence of ICL.
- Comprehensive and thought-provoking discussion, posing clear and relevant questions: discussion of future research directions and acknowledgment of the limitations of the conducted experiments. It identifies the need for experiments on large language models, exploration of other datasets and context lengths, and investigation of additional mechanisms in transformers, encouraging further advancements in the field.
- Consideration of the impact of training duration and stopping criteria: The paper emphasizes the importance of appropriate training duration and stopping criteria to preserving ICL. It suggests that training large transformer models beyond a certain point may result in the gradual disappearance of ICL, even if training losses continue to decrease. This insight has practical implications for model development and training practices.

Research Problem and Motivation:
- The research problem is well-defined and clearly stated, focusing on the transience of ICL in transformer networks.
- The paper provides a convincing motivation for the chosen problem by highlighting the importance of evaluating appropriate validation metrics, preserving ICL, and understanding the factors influencing its persistence.
- The relevance and importance of the research problem are clearly explained, emphasizing the potential impact on training large-scale language models.
- The paper raises numerous additional questions, stimulating further scientific inquiry and exploration.

Methodology:
- The methodology is clearly described and well-founded. The paper outlines the experimental setup, including the models, training procedures, and evaluation metrics used.
- The algorithms, models, and techniques employed are appropriate for addressing the research problem of studying ICL transience in transformer networks.
- The paper adequately explains the experimental setups and data collection processes, providing sufficient details for reproducibility.

Experimental Evaluation:
- The experimental results are presented clearly and concisely, with appropriate figures and tables.
- The evaluation methodology is rigorous and appropriate, considering various metrics and comparing experimental conditions.
- While the statistical significance of the experimental results is not explicitly mentioned, the paper conducts a thorough analysis and provides empirical evidence to support the presented findings.
- The rigorous analyses demonstrate a commitment to scientific rigor and methodology.

Related Work:
- The paper comprehensively reviews the relevant literature, discussing previous works on ICL, language models, optimization, and initialization methods.
- The paper differentiates itself from existing works by focusing on the transience of ICL and investigating strategies to mitigate it.
- The references and citations are appropriate and up-to-date, incorporating relevant studies in the field.

Clarity and Organization:
- The paper is well-structured with clear sections and subheadings, making it easy to follow the flow of ideas.
- The ideas and concepts are presented logically and coherently, building upon each other to support the main arguments.
- The language used in the paper is clear and concise, minimizing excessive technical jargon and enhancing readability.

Presentation and Visualization:
- The plots are visually appealing, utilizing scalable vector graphics and an attractive gradient color scheme. The figure captions are concise and informative.
- The writing is of high quality, with minimal errors or inconsistencies.
- The overall presentation is visually appealing and well-designed, enhancing the readability and comprehension of the paper.

**Weaknesses:**

- Lack of experiments on large-scale datasets: The experiments mainly focus on controlled synthetic datasets, and the impact of ICL transience on real-world, large-scale datasets is not explored. Including experiments on diverse, natural language datasets would enhance the practical relevance of the findings.

- Limited scope of experiments: The experiments were not performed on large language models, one of the most important applications of transformers. Conducting experiments on larger models would provide a more comprehensive understanding of the phenomenon and its implications.

- Narrow definitions of ICL and IWL: The definitions of in-context learning (ICL) and in-weights learning (IWL) used in the paper are relatively specific and make for a clear-cut distinction that in reality - as acknowledged in a footnote - may not be that clear. Exploring additional mechanisms and interactions between context and learning could contribute to a more comprehensive understanding of transformer models.

- Limited exploration of alternative explanations: The paper acknowledges possible explanations for the emergence and transience of in-context learning but does not thoroughly explore alternative hypotheses. Further investigation into optimization quirks, initialization schemes, or alternate optimizers could provide deeper insights into the phenomenon.

Overall, the above-mentioned points can be seen more as ideas for future work rather than genuine weaknesses of the paper.

Minor points:
- Lack of statistical significance analysis: Although the experimental results are presented clearly, the paper does not explicitly address the statistical significance of the findings. Including statistical tests or discussing the significance of the observed differences would strengthen the conclusions.

- (minor) Maybe explain in-weight learning in the abstract, contrast it to in-context learning
- (minor) second to last sentence in the abstract (l. 13), reformulate: how does that question logically follow from the observation?
- (minor) l. 18 Reformulate definition to be clearer, order of words
- redo the references, missing information (ls. 373, 394, 408), inconsistent formatting (e.g., ls. 307, 315)
- Capitalize "In-Context" instead of "In-context" in the title.
- Clearer motivation for why researching ICL is interesting and relevant?


**Questions:**

- Were the plots in the paper based on multiple or single runs? If only a single run was used, how was statistical significance ensured?
- Have you considered the potential interplay between ICL and other important mechanisms in transformers, such as contextual information used in conjunction with IWL for narrowing down possible outputs via copying?
- Figure 5 shows how many runs were averaged, and what do the error shadings indicate?
- Could you further explain the statement that the balance between ICL and IWL is a mirror reflection of grokking? Why would ICL, being more generally useful, be transient instead of persisting?
- Have you explored alternative regularization techniques or approaches to achieve the coexistence of ICL and IWL?
- When the regularization value exceeds 1e-4, ICL transience returns and IWL also does not work. Does this imply a decrease in accuracy with more training? What is the relationship between regularization, ICL, and IWL?
- What may be the underlying mechanism by which regularization affects the persistence of ICL? Does it make IWL more difficult or even impossible, leaving only ICL as a viable option?
- How do you envision your findings' potential real-world applications or implications on ICL and its transience in transformer networks?
- How might this research impact the development and training of AI models in practical scenarios?

**Limitations:**

While the potential negative societal impacts are not explicitly discussed in the provided excerpt, the authors have identified areas for further investigation and improvement in their research, which addresses some limitations of their work.

Since this work focuses primarily on analytical rather than directly enhancing capabilities, there are no immediate negative effects to be considered.

---

> ### Author Rebuttal · Authors · 2023-08-10
>
> We thank you for your thoughtful and impressively thorough review! We have factored in the suggested minor edits, and we address the questions below:
>
> Q1: Plots in the paper were based on two runs. We observed little variance between these two runs (as shown by the lighter colored error bars around curves – these are often hard to see because of the low variance!), and thus decided to not do more random seeds, given the significant cost of each run. Nonetheless, we observed ICL transience across many runs when compiled across conditions, and this replicability of ICL transience across conditions indicates robustness of the overall phenomenon.
>
> Q2: Yes! We believe this to be a possible mechanism at work in our models, and plan to investigate it further. However, we'd like to emphasize the focus of the present work is on the transience of ICL.
>
> Q3: Error shadings indicate standard error of the mean.
>
> Q4: By mirror reflection here we mean that ICL transience is the *opposite* of grokking. In grokking, networks transition from less general to more general circuits, while ICL transience appears to be the opposite (assuming ICL is viewed as "more general") – networks transition from ICL solutions to IWL solutions. We will reword this part of the text so that it is clearer.
>
> Q5: Not yet, but this is an exciting avenue for future work.
>
> Q6: Above that scale (our run in the appendix with 1e-3), the regularization is too high for the network to retain either circuit – if we train for long enough, the network will lose ICL and IWL, as indicated by the train loss plateauing at a higher value (Figure 11).
>
> Q7: That is what we believe to be happening, though a mechanistic analysis (with ablations of subcircuits) would likely be necessary to verify. To further investigate these results, we have performed an additional analysis, applying regularization to different parts of the network (Figure 3, attached page). We find that regularization mitigates ICL transience only when it is applied to the dense layers, and not to the attention layers.
>
> Q8-9: We hope our findings will encourage model developers to be more conscious of evaluations of in-context learning throughout training, as compared to just watching the train loss. Given the trend of overtraining models for longer and longer (Touvron et al. 2023), we believe this especially relevant.

---

### Official Review · Reviewer_wdqh · 2023-07-06

**Soundness:** 3 good
**Presentation:** 4 excellent
**Contribution:** 3 good
**Rating:** 6
**Confidence:** 4

**Summary:**

The paper investigates the emergence and persistence of in-context learning (ICL) in Transformer models. Training Transformer models on tasks by Chan et al. [10], the authors find that ICL does not persist, despite an initial emergence. Loss reduction suggests a preference for in-weight learning after some training steps. Through specialized experiments, the authors explore model size, training dataset statistics, and L2 regularization to understand and mitigate this phenomenon. The findings emphasize the importance of L2 regularization in Transformers during extended training. This research provides valuable insights into ICL dynamics, contributing to better understanding of this phenomenon and offering insights into potential improved training strategies.

**Strengths:**

1. The paper builds upon the established dataset by Chan et al. [10], ensuring reproducibility of results. The phenomenon of interest is consistently observed across different scenarios, allowing for comprehensive examination. The inclusion of small-scale experiments facilitates further investigation by researchers. The availability of the code, given that it will be released, will help with reproducibility and transparency of the study.
2. The authors adopt a systematic approach to address the problem at hand, systematically eliminating various factors that contribute to the lack of persistence in in-context learning (ICL). Through controlled experiments, they explore and analyse multiple aspects that may impact ICL persistence.
3. The paper has a well-structured and coherent storyline, ensuring clarity of the results. The authors effectively motivate each experiment, providing a clear rationale for their choices. I enjoyed reading the narrative presented by the authors.

**Weaknesses:**

1. The phenomenon is observed for a single small dataset. Although some experiments are made with respect to aspects of the training data, i.e. number of classes and number of exemplars per class, the small relative size of the training data in conjunction with the large number of training steps tested for here, may potentially lead to conclusions that do not generalize past this setting. Some attempts were made to match data distribution with naturalistic data, but any conclusion that generalize to large language models have to be taken with a grain of salt.
2. Although a lot of different effects are explored and some correlations explored, no clear answer is given in the end.

**Questions:**

1. Connections to Grokking are not obvious to me. In Grokking [27] memorization at early phase of training is replaced by more general solutions at later stages of training. In your context, the opposite effect seems to be happening, if you assume that ICL are the more general solutions.
2. What is the effect of different positional encodings to the emergence and persistence of ICL? It would be interesting to study.
3. In Figure 7 what is the "chance" prediction for IWL? Different runs here have different number of classes.
4. You can match the color scheme between Figures 9,10 and Figure 11.
5. Line 242-244 "Furthermore, these results imply that it may be desirable to stop training large transformer models before a compute budget is reached—to preserve ICL—even if training losses are still decreasing.": the statement is a bit too bold given the experimental setup of the paper.
6. Line 258-259 "lottery tickets contained in common initialization methods [27, 37] that incentivize ICL": I find this a very interesting conjecture that could be explored.
7. Another potential effect that could be explored may be the "softness" of the softmax function. Perhaps the network realizes that to decrease the loss further it needs to resolve to IWL. It would be interesting to track the entropy or "hardness" of the attention mechanism. If the softmax is deemed to be the deciding factor, you can think about replacing the softmax with a harder version e.g. [1, 2, 3] and see if the transient effect disappears.

[1] Martins, André, et al. "Sparse and continuous attention mechanisms." Advances in Neural Information Processing Systems 33 (2020): 20989-21001.

[2] Hahn, Michael. "Theoretical limitations of self-attention in neural sequence models." Transactions of the Association for Computational Linguistics 8 (2020): 156-171.

[3] Chiang, David, and Peter Cholak. "Overcoming a theoretical limitation of self-attention." arXiv preprint arXiv:2202.12172 (2022).

**Limitations:**

The paper discusses a limited scenario where ICL is not persistent. Experiments are performed but always in the context of the provided dataset and for smaller models. More experiments are in general required to verify the existence of this phenomenon elsewhere. The study is nonetheless very interesting and useful for the community.

---

> ### Author Rebuttal · Authors · 2023-08-10
>
> We thank you for your insightful comments. You have clearly thought carefully about the paper, and we very much appreciate it.
>
> W1: As noted in the general rebuttal, we are running experiments with modifications of the dataset, including running on more symbolic data (Figure 1, attached page). We would also like to note that the data is not actually that "small" in size due to the combinatorial construction of our sequence prediction task (17 input tokens -> one output token). Specifically, there are over 1600 * 1599 * (1598 choose 2) * (8! / (3! * 3!)) * (20^9) = 1.87 * 10^27 possible training sequences!
>
> W2: We will update the paper to emphasize evidence we do provide for mechanisms behind ICL transience – specifically, competition in the residual stream (lines 143-151) – as well as mitigations and what they might indicate about underlying mechanisms (e.g., Figure 3, attached page). We also believe that raising questions and initial evidence is just as important to the process of science as taking the final steps to reach the final answers.
>
> Q1: Yes indeed! This was our intended meaning with  "ICL transience is a mirror reflection of the grokking phenomenon" – we will make this clearer with different wording, that it seems to be the opposite of grokking. This was the observation that motivated our use of weight decay (to try and retain the more "generalizing" ICL solution).
>
> Q2: This is definitely an exciting direction for future work. We ran some preliminary experiments in the rebuttal period on a smaller, 2-layer model and found that positional embeddings can lead to differences in peak height and decay slope. ICL remains transient across settings where it emerges (Figure 4, attached page).
>
> Q3: Good question – the chance predictions here would be 1/# classes (so 1/12800 for the purple line, and 1/1600 for the other two). Due to the scaling of the plot (which we wanted to keep as 0-1 for consistency), we just included the 1/1600 chance line. We will add extra clarification to the text and image captions.
>
> Q4: Thank you for pointing this out. We will make this change.
>
> Q5: We will soften that statement – thank you for the recommendation! We will modify it to the following (please let us know if you recommend further changes): "Furthermore, these results suggest that it may be desirable to stop training some transformer models before a compute budget is reached—to preserve ICL—even if training losses are still decreasing. This would need to be ascertained on a case by case basis. At the least, evaluations should be performed to ensure that ICL performance remains within desired levels throughout training."
>
> Q6: We find this very interesting as well, and hope to explore this idea in future work!
>
> Q7: Agreed! We thank you for the additional references, which we will add to our discussion of the "imperfect match" theory (lines 266-273). We are also excited to try replacing the attention mechanism in future work, as you suggest.

---

> > ### Comment · Reviewer_wdqh · 2023-08-15
> >
> > Thank you for the additional details.
> >
> > Interesting to see that longer contexts don't seem to affect the emergence of this behaviour. I appreciate the authors for experimenting with different positional encodings. Results seem to indicate that positional encodings may affect the transient nature of ICL in Transformers. More investigation will clarify this in the future.
> >
> > Overall, I find the work a valuable addition towards understudying generalization in Transformers. Some of the results are preliminary and empirical evidence could be expanded. Nonetheless, the overall presentation is excellent. Hope this might inspire future work! I will keep my original score.

---

### Official Review · Reviewer_iyV5 · 2023-07-06

**Soundness:** 3 good
**Presentation:** 3 good
**Contribution:** 3 good
**Rating:** 6
**Confidence:** 3

**Summary:**

The paper explores In-Context Learning (ICL), a model's ability to adapt its behavior during inference without weight updates, and In-Weights Learning (IWL), where the model relies on information stored in its weights. While early neural networks displayed ICL in controlled scenarios, research changed with the transformer model, particularly with GPT-3, which exhibited ICL without deliberate training efforts. However, it was found that ICL isn't guaranteed when training transformers, with language data properties like burstiness and skewed distribution playing a crucial role. Additionally, ICL and IWL often seem to oppose each other, with ICL emerging more readily when training data has a larger number of tokens or classes and is bursty. Despite the challenges of relying on large models, overtraining is currently a favored approach for training small yet high-performing transformers. The paper challenges the assumption of persistence (that once ICL emerges it remains), showing that ICL can be transient, with the potential to disappear if networks are trained carelessly for longer periods.

**Strengths:**

On the positive side, the paper is certainly quite interesting and relevant to the major open question these days, which is the emergent capability of large language models. The paper raises a number of interesting open questions (see Discussion in section 7). In fact, I would argue that that is the most interesting section of the paper.

**Weaknesses:**

On the negative side, the paper can be viewed as "half-baked" at this point. It relies heavily on some empirical results, derived mostly from small models and visual tasks, and it raises a number of questions that are interesting but.. open. I wish the paper could go a bit more deeply to actually answer at least one of those questions. At this point I cannot say with confidence that the paper is ready to be presented at the premier conference of deep learning.

**Questions:**

My major question is -- is the process of Chan et al. [10], which has been adopted by the authors of this paper, really capable to answer such fundamental questions about ICL and the emergent capabilities of LLMs? The queries that the paper relies on very short -- just 8 exemplar-label pairs and one query image -- i.e., just a sequence of length 17. That also means that the provided "context" is very short. this could be a reason that ICL is not persistent.

Another potential issue is the "context" in these experiments is very highly informative regarding the given query. This is why we see that ICL gives a much much larger accuracy than IWL in all plots of the paper. Is this a realistic/pragmatic evaluation regime? I would assume that in practice the context is only partially informative (at least for NLP tasks) and that the network has to rely in anyway much more on trained weights.



**Limitations:**

The paper is quite open about its limitations, mentioning clearly that it has not utilized very large models and so it is not really certain whether the major claims of this study would be actually still seen in large models such as GPT3 or 4. Additionally the paper does not try to "oversell" what it does -- and it identifies many open questions instead of claiming that it already has answers to those questions.

---

> ### Author Rebuttal · Authors · 2023-08-10
>
> We appreciate the thoughtful feedback, and are glad you were excited by the questions our work raises. We believe that answering each of these questions is a difficult endeavor in itself, and that answering them is a challenge for the whole community that will happen incrementally. Our work takes the crucial first steps by documenting the phenomenon of ICL transience, providing some explanation (lines 143-151, competition in the residual stream), and providing hypothesis-driven mitigations (Section 5, weight decay). We have also performed new experiments that provide additional convergent evidence that ICL transience is driven by competition with IWL circuits (Fig 3, attached), which we will add to the paper. Finally, we end with an honest discussion of open questions, which we believe will motivate future work.
>
> In response to your question about context length, we conducted some short runs with twice the context length (Figure 2, attached page) – 16 image-exemplar pairs (for a sequence length of 33), 6 of which come from the query class, 6 from a distractor class, and 4 random. We had difficulty getting ICL to emerge consistently in this setting, across 4 seeds, only 2 showed above chance ICL at all (both with a lower peak height). Consistent with our experiments though, in both cases where ICL appeared, it was transient. We believe this longer context might actually be less informative, perhaps due to a more "imperfect match" (one of the theories we suggest for ICL transience, lines 266-273) given the larger number of distractor exemplar-label pairs, or recency biases known to exist in transformers due to the fixed sinusoidal positional embedding scheme.
>
> With regards to the "highly informative" context of our original experiments, we believe this makes the transience of ICL even more surprising! If ICL fades even with such useful context, we believe it likely that overtraining with less informative contexts, all else the same, would also lead to transience.

---

> > ### Comment · Reviewer_iyV5 · 2023-08-10
> >
> > thank you for the thoughtful rebuttal. After also considering the other reviews, I have decided to stay with my original evaluation score.

---

### Official Review · Reviewer_oS5Q · 2023-07-14

**Soundness:** 2 fair
**Presentation:** 2 fair
**Contribution:** 2 fair
**Rating:** 5
**Confidence:** 2

**Summary:**

This paper presents empirical evidence illustrating the transient nature of in-context learning (ICL) in transformers, and it shows that the models may shift to in-weights learning (IWL) when overtrained. Furthermore, the transience of ICL is studied across diverse settings such as model size, dataset size, data distributions, and training regularization. Results reveal that adequate L2 regularization can mitigate this transience.

**Strengths:**

1. This paper reveals a characteristic of transformers performing transient ICL.
2. The proposition that regularization can eliminate the transience is noteworthy.
3. The provided experiments are clearly organized and easy to follow.

**Weaknesses:**

1. The paper lacks clarity in its notations, which makes them hard to follow without reference to previous literature. For instance, the term "in-weights learning" needs to be related to supervised learning; in the Zipfian distribution equation $p(X = x) \propto x^{-\alpha}$, the terms $X$ or $x$ should be explained as representing the rank of input classes.
2. While the authors provide numerous empirical evidences, they scarcely discuss potential underlying reasons. Some interpretations or conjectures would enrich the discussion around the observed outcomes.
3. The experiments are constrained to a single dataset and transformer (with different sizes), which is not convincing enough in my opinion. It would be interesting to see if the similar phenomenon is observed with other datasets (e.g., NLP dataset) or other auto-regressive models. The usage of the Omniglot dataset—a stationary supervised dataset—where in-context samples can be regarded as independent entries, makes the existence of IWL unsurprising.

**Questions:**

1. Why are the IWL accuracies consistently low? Would further training iterations increase the IWL accuracy and what is the maximum achievable IWL accuracy?
2. Have the authors considered that the observed ICL performance drop might be due to the fact that evaluation in-context data distribution is different from training, especially as the label space changes? In Fig 5(c), when evaluation and training use the same data distribution, the performance does not decay.
3. The connection between Fig5(a)(b) and Fig5(c)(d) seems unclear given the different training datasets. What's more, in Section 4.1, the authors' claim "Figure 5d shows that larger embedding dimensions enhance the model’s ability to perform IWL, but do not consistently help ICL (Figure 5c)." seems puzzling. Since ICL accuracies are already close to ones, is there room for improvement? Would reducing the embedding dimension identify a threshold beyond which ICL accuracy is $<1$?
4. In the Discussion section, the authors note "However, after training for a long time, we find that even rare classes begin to be learnt in-weights." However Section 4.3 does not provide evaluation over rare classes.

**Limitations:**

yes

---

> ### Author Rebuttal · Authors · 2023-08-10
>
> Thank you for your thoughtful feedback and questions! Please find responses to your comments below:
>
> W1: We believe in maximizing the clarity of our work, and appreciate your suggestions for doing so. We will clarify the definition of Zipfian distributions. We will also add additional explanation regarding the relationship between "in-weights learning" and supervised learning.
>
> W2: We agree that these novel results provoke interesting new questions about the mechanisms of in-context learning in transformers. The experiments on embedding size provide a clue as to why we observe ICL transience (lines 134-151) – we carefully designed those experiments to help us see that IWL and ICL circuits are in competition for capacity in the transformer. We have also run additional experiments elucidating how and why weight decay mitigates transience (see general rebuttal and Figure 3, attached page) – these results provide additional evidence for ICL-IWL competition as a mechanism driving ICL transience. We will update the paper with these results, and emphasize this takeaway more clearly in the paper. We also give some further conjectures on why ICL is transient in the Discussion (lines 253-274), though we refrain from over-speculating.
>
> W3: The scope of our paper is focused on transformers, given that ICL has very different mechanisms and behavior in transformers vs RNNs (e.g., as found in Chan et al. (2022)). We do not make any claims that ICL transience extends beyond transformers, and we will emphasize this in the paper. With regards to datasets, we extended to fixed embeddings (Figure 1, attached page) and have started runs with one-hot data and Gaussian in-class variation. We maintain a paired exemplar-label setup as it allows for careful variation and controls (see general rebuttal and attached page). Such synthetic approaches are common in the study of ICL, as noted in our related work (lines 208-214). Extending to uncontrolled, web-scraped natural language is an exciting future direction, but is beyond the scope of our current work.
>
> By design, our training task can indeed be solved quite naturally by IWL, but also by ICL. The critical insight of our work is that the IWL and ICL solutions trade off in surprising ways over the course of training, even on this “stationary” dataset – specifically, ICL emerges and then fades away. This behavior has not been previously reported.
>
> Q1: We believe further iterations would continue to increase IWL accuracy, as the slope is positive and shows no sign of plateauing. IWL accuracy is low as the IWL eval sequences are out-of-distribution (purposefully) for the model. For example, models are known to exhibit naive copying biases (as noted by reviewer 8T18, and Elhage et al. (2021)), which are no longer useful on the IWL sequences, which could be contributing to the lower accuracy – we note this in Footnote 2 (page 2).
>
> Q2: The paper focuses on in-context learning that is *emergent*, i.e. which appears even when the training data does not require in-context learning as a solution (and therefore where it is out of distribution with respect to the training data). This emergent phenomenon is observed widely in transformers (e.g., Brown et al. (2020), Chan et al. (2022)). The results in Figure 5(c) are only meant to demonstrate that the in-context learning task is "solvable". Our main claim is that transformers, when trained on data that *does not* require ICL, first learn to do ICL (corroborating others' findings of emergent ICL), but then subsequently lose ICL abilities, which may have implications for the current trend of training transformer models for longer and longer (c.f. Touvron et al. 2023).
>
> With regards to the specific 0-1 label space, we'd conducted early experiments across different random label spaces and found only a small amount of variance. Specifically, across 11 additional random relabelings (in addition to the 0-1 relabeling), we calculated the 95% confidence interval to be +/- 2% around the peak, and +/- 1% on the decay slope. Given these, we opted for the simpler, more reproducible setup of using just the 0-1 relabeling. We will update the paper to explain this choice.
>
> Q3: In Figures 5(a)(b), we find that larger embedding dimensions mitigate ICL transience. Figure 5(c)(d) are follow-up experiments that are carefully designed to disambiguate between two possible explanations for this finding. Specifically, we argue that ICL is not inherently worse at smaller embedding dimensions (as seen in 5c), so the mitigated transience (seen in 5a) is likely due to less competition to IWL circuits. There is likely an embedding size threshold below which ICL does not emerge, but our main purpose in providing these figures was to suggest a possible explanation – competition in the residual stream – for why ICL may be transient. We will clarify the paragraph where we make these points (lines 143-151).
>
> Q4: The overall increase in IWL accuracy led us to believe that accuracy was increasing across classes, but we agree that this isn't necessarily the case and thank you for pointing this out. We will look into the specific IWL accuracies for classes according to rarity, updating Section 4.3 and this part of the discussion accordingly.
>
> Thank you once again for your detailed comments and feedback. We hope that we’ve managed to address your concerns. If not, we’d be more than happy to continue the discussion!

---

> > ### Comment · Reviewer_oS5Q · 2023-08-20
> >
> > Thank you for your detailed response and the additional experiments provided. I have a few suggestions for the updated version:
> >
> > 1. Could you investigate if training the transformer for more iterations would lead the model to perform IWL precisely?
> > 2. I'm curious if the observed phenomenon - where ICL emerges first, then fades, giving way to IWL - is due to the dataset's classification difficulty. How would the results differ if the dataset had fewer classes, say 10 or even less?
> > 3. Could you provide evidence that, after more iterations, the rare classes of Zipfian data are indeed learned in weights?
> >
> > A minor suggestion on presentation: Enhance the clarity of your plots, perhaps by rescaling the y-axis (e.g., Fig 9(b)), to better highlight IWL performance.
> >
> > I've decided to increase my score from 4 to 5, and I am inclined towards acceptance.

---

> > > ### Author Response · Authors · 2023-08-20
> > >
> > > Thank you for your remaining comments, and your recommendation for acceptance!
> > >
> > > 1. We are indeed performing extra long runs to investigate the behavior on IWL evaluation in the long run, partly motivated by some of your questions in the initial review. The results are coming in slowly, but the IWL evaluation has continued to show gradual improvement -- we can definitely include these results when the runs complete.
> > >
> > > 2. It is a good question re number of classes. As found in Chan et al. (2022), we have also found that a given level of difficulty is necessary for ICL to emerge in the first place. For example with 10 classes, the model never learns ICL, and jumps directly to the IWL solution. We believe that this is because the IWL is too much easier than ICL, in that scenario. These results replicate the interesting results of Chan et al. (2022).
> > >
> > > 3. We are unfortunately still running these newer Zipfian experiments, so we cannot report on the results. For now, we are removing that statement from the paper. Thank you so much for directing our attention to this question -- we will only re-incorporate that statement once we have definitive evidence from the new experiments.
> > >
> > > Thank you also for your suggestion regarding the presentation of the plots.

---

### Author Rebuttal · Authors · 2023-08-10

We thank the reviewers for their detailed feedback and questions. Their comments and suggestions have enabled us to make significant improvements to the work.

We are glad the reviewers found the work  "interesting and relevant to the major open question these days" and that they enjoyed the "clarity of the results" and "comprehensive and thought-provoking discussion, posing clear and relevant questions". We believe that our work contributes significantly to the scientific body of work around understanding transformers by (a) identifying an interesting new phenomenon (ICL transience), (b) providing initial explanations as to why it arises (Section 4.1), and (c) giving a hypothesis-driven solution for how to mitigate ICL transience (Section 5).

We share the reviewers' enthusiasm to see these results in LLMs. Given our resource restrictions, we opted for the data generating process of Chan et al. (2022), prioritizing more controls and experimental conditions, as compared to limited large-scale runs on uncontrolled internet-scale language data. While our work does not directly consider LLMs, we hope it will convince large-scale practitioners to consider the possible transience of ICL, which may motivate more careful evaluations of ICL throughout training.

We also share the reviewers' desire to understand the mechanism behind this interesting phenomenon. Our existing results suggest that as IWL increases, ICL gets crowded out due to competition in the residual stream (lines 134-151). We have conducted additional experiments with weight decay applied selectively to MLP layers of the network (which Geva et al. (2021) indicate may be primarily responsible for in-weights information). We found that in these cases, IWL does not arise, and ICL persists (red line, Figure 3, attached page). We believe these results provide compelling observational evidence that ICL fades due to competition with IWL circuits, and will emphasize this in the paper. Our work documents a hitherto unknown phenomenon and also takes the crucial first steps towards elucidating underlying mechanisms, which we hope will spur further inquiry in the academic community on the theories governing in-context learning.

To address reviewers' questions, we present additional experimental results in our one page report, which we will also incorporate into our paper. To this end, we have extended to a dataset with fixed, rather than learned, embeddings (Figure 1). We also conducted experiments with longer context lengths (Figure 2, reviewer iyV5) and experiments on the effect of position embeddings (Figure 4, reviewer wdqh).

---

### Author Response · Authors · 2023-08-20

Thank you so much to the reviewers for all your detailed comments and questions. We believe that this discussion period has substantially improved both the content and clarity of our paper. Thank you also for your unanimous recommendations for acceptance -- we are honored and would be grateful to share our work at NeurIPS this year.

---

### Decision · Program_Chairs · 2023-09-21

**Decision:**

Accept (poster)

**Comment:**

The paper presents new empirical understandings of in-context learning (ICL) in transformers. The main finding is that the transformer shifts its mechanism from ICL to in-weights learning (IWL), when the pretraining data can be solved by both mechanisms, and the transformer is trained for long enough. The paper also provides ablations and proposes mitigation strategies.

The opinions about this paper are mixed. The reviewers have concerns about the setting being a bit too specific, but agree that the message is interesting and worthy to share with the community. After further discussions, the reviewers are supportive of this paper and are optimistic about the discussions it would raise, despite the concerns. Therefore, I recommend acceptance.

I encourage the authors to incorporate the additional results (in rebuttal) and the reviewers' suggestions into the revision, to further improve the paper for the publication.